# Real-space imaging of confined magnetic skyrmion tubes

M. T. Birch [1,2], D. Cortés-Ortuño [3], L. A. Turnbull[1], M. N. Wilson[1], F. Groß [4], N. Träger[4], A. Laurenson [5], N. Bukin[5], S. H. Moody[1], M. Weigand [4,6], G. Schütz[4], H. Popescu[7], R. Fan[2], P. Steadman[2], J. A. T. Verezhak[8], G. Balakrishnan [8], J. C. Loudon[9], A. C. Twitchett-Harrison[9], O. Hovorka[3], H. Fangohr [3,10], F. Y. Ogrin[5], J. Gräfe [4] & P. D. Hatton [1✉]

Magnetic skyrmions are topologically nontrivial particles with a potential application as information elements in future spintronic device architectures. While they are commonly portrayed as two dimensional objects, in reality magnetic skyrmions are thought to exist as elongated, tube-like objects extending through the thickness of the host material. The study of this skyrmion tube state (SkT) is vital for furthering the understanding of skyrmion formation and dynamics for future applications. However, direct experimental imaging of skyrmion tubes has yet to be reported. Here, we demonstrate the real-space observation of skyrmion tubes in a lamella of FeGe using resonant magnetic x-ray imaging and comparative micromagnetic simulations, confirming their extended structure. The formation of these structures at the edge of the sample highlights the importance of confinement and edge effects in the stabilisation of the SkT state, opening the door to further investigation into this unexplored dimension of the skyrmion spin texture.

[1] Centre for Materials Physics, Durham University, Durham DH1 3LE, UK. [2] Diamond Light Source, Didcot OX11 0DE, UK. [3] Faculty of Engineering and Physical Sciences, University of Southampton, Southampton SO17 1BJ, UK. [4] Max Planck Institute for Intelligent Systems, 70569 Stuttgart, Germany. [5] School of Physics and Astronomy, University of Exeter, Exeter EX4 4QL, UK. [6] Helmholtz-Zentrum Berlin für Materialien und Energie GmbH, Institut Nanospektroskopie, Kekuléstrasse 5, 12489 Berlin, Germany. [7] Synchrotron SOLEIL, Saint Aubin, BP 48, 91192 Gif-sur-Yvette, France. [8] Department of Physics, University of Warwick, Coventry CV4 7AL, UK. [9] Department of Materials Science and Metallurgy, University of Cambridge, Cambridge CB3 0FS, UK. [10] European XFEL GmbH, Holzkoppel 4, 22869 Schenefeld, Germany. ✉email: p.d.hatton@durham.ac.uk

Magnetic skyrmions have seen a flurry of recent research interest as potential information elements in future spintronic device architectures due to their particle-like, topologically protected nature[1,2] and current-induced mobility[3]. Skyrmion states are typically stabilised by the interplay of the ferromagnetic exchange and Zeeman energies with the Dzyaloshinskii-Moriya Interaction (DMI)[4]. In ferromagnet/heavy metal multilayer thin films, interfacial DMI is induced by symmetry-breaking spin-orbit coupling at the interface between the layers, leading to the formation of Néel-type skyrmions[5–8]. Bulk DMI, arising due to the lack of centrosymmetry in the underlying crystal lattice, is responsible for the formation of Bloch-type skyrmions in a range of chiral ferromagnets[9–13]. In crystals of these bulk materials the skyrmion state is typically only at equilibrium in a limited range of applied magnetic field and temperature just below the Curie temperature, $T_C$, forming a hexagonal skyrmion lattice (SkL) in a plane perpendicular to the applied magnetic field.

The three dimensional visualisation in Fig. 1 depicts the extended spin structure of three magnetic skyrmion tubes. The dynamics of this skyrmion tube (SkT) state play an important role in the creation and annihilation of skyrmions, and have potential applications in magnonics-based computing[14,15]. For example, metastable skyrmions, which are created beyond the equilibrium thermal range by rapid field cooling[16,17], are thought to unwind into topologically trivial magnetic states through the motion of a magnetic Bloch points. When transitioning to the helical state, such Bloch points are thought to zip together neighbouring skyrmion tubes[18]. On the other hand, when transitioning to the conical state, it has been suggested that skyrmion tubes unwind via the motion of Bloch points formed at the end of each individual tube[19,20], as depicted in Fig. 1.

Real-space observation of the vertical dimension of the SkT state and these associated dynamics requires an in-plane magnetic field applied perpendicular to the imaging axis. Electron imaging techniques such as Fresnel Lorentz Transmission Electron Microscopy (LTEM)[10,11], and electron holography[21,22] have been widely utilised to image magnetic skyrmions. However, due to the deflection of electron trajectories by magnetic fields, these techniques do not easily allow for the application of an in-plane magnetic field[23].

In magnetically sensitive X-ray based techniques, such as X-ray holography and Scanning Transmission X-ray Microscopy (STXM), the probe particles are not deflected by magnetic fields, and therefore imaging with an in-plane applied magnetic field is feasible. X-ray techniques possess further advantages including the possibility to reconstruct three dimensional magnetic structures using vector field tomography[24], and a picosecond time-resolution capable of probing skyrmion dynamics[25–27]. However, X-ray holography has seen only limited use for imaging bulk DMI skyrmions[28], while STXM instruments have previously lacked cryogenic temperature capabilities, limiting their application to observing interfacial DMI skyrmions in multilayer thin films[7,8].

In this work, we utilise both X-ray holography and cryogenic STXM to image chiral spin textures in FeGe lamellae, and, with comparative micromagnetic simulations, demonstrate the real-space observation of magnetic skyrmion tubes. The results highlight the importance of confinement effects in stabilising the SkT state under an in-plane applied magnetic field, paving the way for future studies of skyrmion tube dynamics.

## Results

**X-ray imaging of chiral spin textures**. Magnetic phase diagrams of a ~120 nm thick FeGe lamella (see Methods) are displayed in Fig. 2a, b for magnetic fields applied out-of-plane and in-plane,

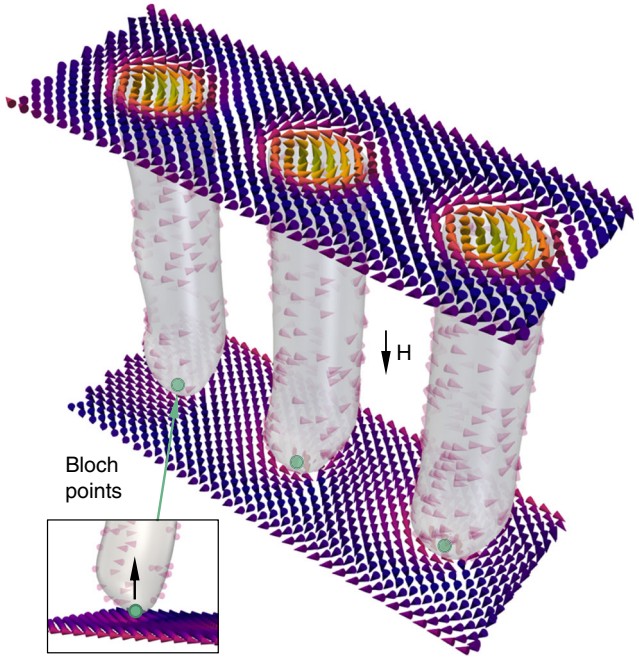

**Fig. 1 Visualisation of the skyrmion tube spin texture.** Three dimensional visualisation of three magnetic skyrmion tubes from the micromagnetic simulations presented in this paper, illustrating their extended spin structure. The inset highlights the location of the magnetic Bloch point at the end of each skyrmion tube.

respectively, as determined by magnetic diffraction measurements (see Methods, Supplementary Figs. 2 and 3, and Supplementary Notes 1 and 3). At low applied magnetic fields the helical state is at equilibrium, consisting of a continuous rotation of spins orthogonal to a propagation vector, as depicted in Fig. 2c. This vector lies in the plane of the lamella along a preferred axis determined by the present cubic anisotropy[29]. Upon application of an out-of-plane magnetic field, the SkL state is formed, illustrated in Fig. 2d. In this field configuration, the extent of the equilibrium SkL state is greatly expanded in temperature and field in comparison to bulk (see Supplementary Fig. 1, Supplementary Note 1). This phenomenon has previously been attributed to shape anisotropy and confinement effects due to the reduced dimensionality of the sample[30–33]. At higher out-of-plane magnetic fields, the magnetisation is expected to form the out-of-plane conical and field polarised states. However, these are indistinguishable for diffraction measurements in this field configuration.

When an in-plane magnetic field is applied, the helical state rotates as it transitions to the conical structure, which is comprised of a continuous rotation of spins at an acute angle to a propagation vector aligned parallel to the applied magnetic field, as shown in Fig. 2e. The application of an in-plane magnetic field is also expected to stabilise the in-plane SkT state. However, we found that in this field configuration the extent of the equilibrium skyrmion region was greatly suppressed in our lamella samples, possibly entirely, as evidenced by the lack of an identifiable SkT state in Fig. 2b. This behaviour can be expected when considering that the effects of shape anisotropy and confinement, which enhance the stability of the SkL in the out-of-plane field configuration, may work to reduce the stability of the SkT state for the in-plane field arrangement. We note that due to the sample construction required for these diffraction measurements, the field of view was limited to the centre of the lamella, and therefore it was not possible to detect potential formation of a

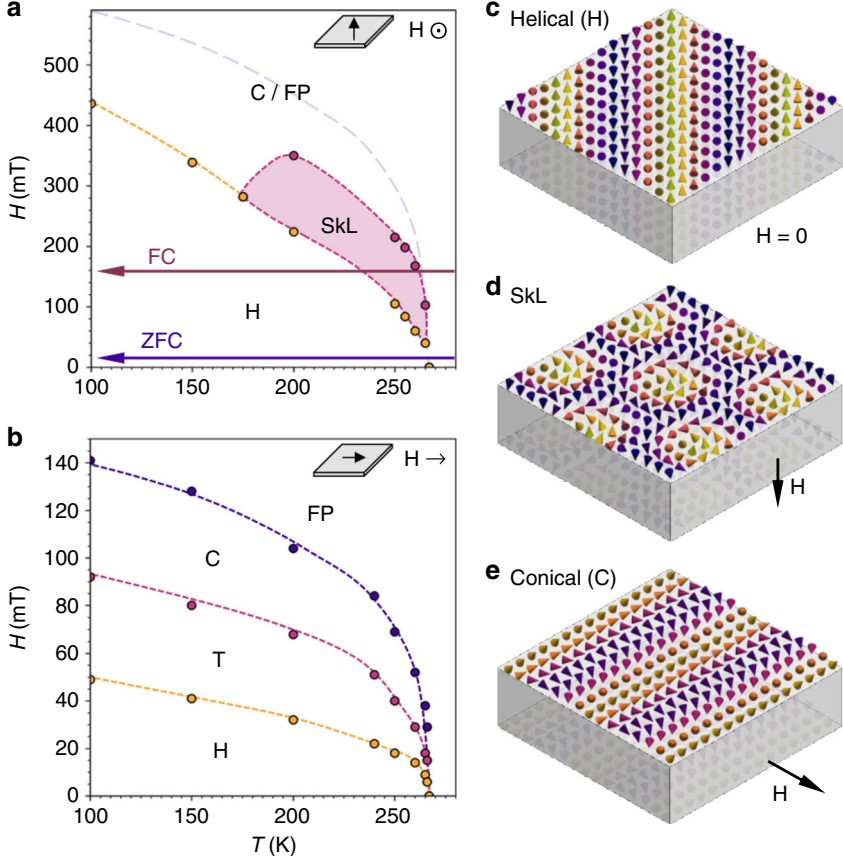

**Fig. 2 Spin textures and magnetic phase diagrams of an FeGe lamella. a, b** Phase diagrams of the ~120 nm FeGe lamella for out-of-plane and in-plane applied magnetic field, respectively, as determined magnetic x-ray diffraction. Schematics of each field configuration are shown as insets. In **a**, boundaries between the helical (H) and skyrmion lattice (SkL) states are displayed by yellow and magenta dots. The expected boundary between the indistinguishable conical (C) and field polarised (FP) states is estimated by the purple dashed line. The zero-field cooling (ZFC) and field cooling (FC) procedures are indicated. In **b**, yellow, magenta and purple dots boundaries indicate the boundaries between the helical (H), helical rotation transition (T), conical (C) and field polarised (FP) states. **c–e** Schematic illustrations of the spin textures as they are expected to appear in a thin lamella.

SkT state at the edges of the sample (see Supplementary Fig. 1, Supplementary Note 1).

Magnetic contrast images acquired by the X-ray holography, STXM and LTEM techniques are presented in Fig. 3. Both X-ray imaging techniques achieve magnetic contrast by exploiting the resonant enhancement of the magnetic scattering and absorption of X-rays close to the $L_3$ absorption edge of the magnetic Fe atoms, exhibiting a signal proportional to $m_z$, the out-of-plane component of the sample magnetisation (see Methods, Supplementary Fig. 4, and Supplementary Note 4). In contrast, LTEM provides the in-plane components of the magnetic flux generated by the underlying magnetisation, highlighting the complimentary nature of these two techniques[34] (see Methods). Simulated X-ray images were created from comparative micromagnetic simulations by averaging the out-of-plane magnetisation $m_z$ through the thickness of the simulated spin texture, and are in excellent agreement with the experimental images (See Methods, Supplementary Figs. 7 and 8, and Supplementary Note 6 for details).

To achieve sufficient magnetic contrast in both the X-ray holography and STXM measurements, we found that it was necessary to maximise the ordered magnetic moment by acquiring images below 150 K. The helical state is featured in Fig. 3a–d, demonstrating the formation of stripe-like structures in diagonal orientations with a measured period of ~70 nm. Figure 3e–h displays images of the SkL state for an out-of-plane applied magnetic field, with a measured period of ~83 nm. As no equilibrium SkL state is present at 150 K and below, we utilised

field cooling to generate a metastable SkL state for the X-ray images presented in Fig. 3f, g. Images of the conical state under an in-plane magnetic field are displayed Figs. 3i–k, with a period of ~70 nm.

**Observation of skyrmion tubes**. After demonstrating successful X-ray imaging of chiral magnetic structures for both out-of-plane and in-plane applied magnetic fields, we investigated the possibility of observing the in-plane magnetic SkT state. When observed perpendicular to its central axis, an individual Bloch skyrmion tube is expected to exhibit both light and dark contrast, as the spins point in opposing directions either side of the central skyrmion core. Figure 4a displays a STXM image acquired after field cooling the second FeGe lamella under an applied in-plane magnetic field of 35 mT. The three pairs of light and dark horizontal stripes at the bottom of the image, which are situated in the corner of the sample (see Supplementary Figs. 1 and 6, and Supplementary Note 5), are aligned along the applied magnetic field direction, and thus have the expected appearance of the SkT spin texture embedded in the vertical stripes of the conical state. While the uppermost skyrmion tube bends directly into the conical stripes, the two lower tubes appear to bulge outward at their ends before terminating in the conical state. Upon increasing the applied magnetic field, the skyrmion tubes decrease in length, before being annihilated by the conical state at 130 mT, as shown in Fig. 4b–d.

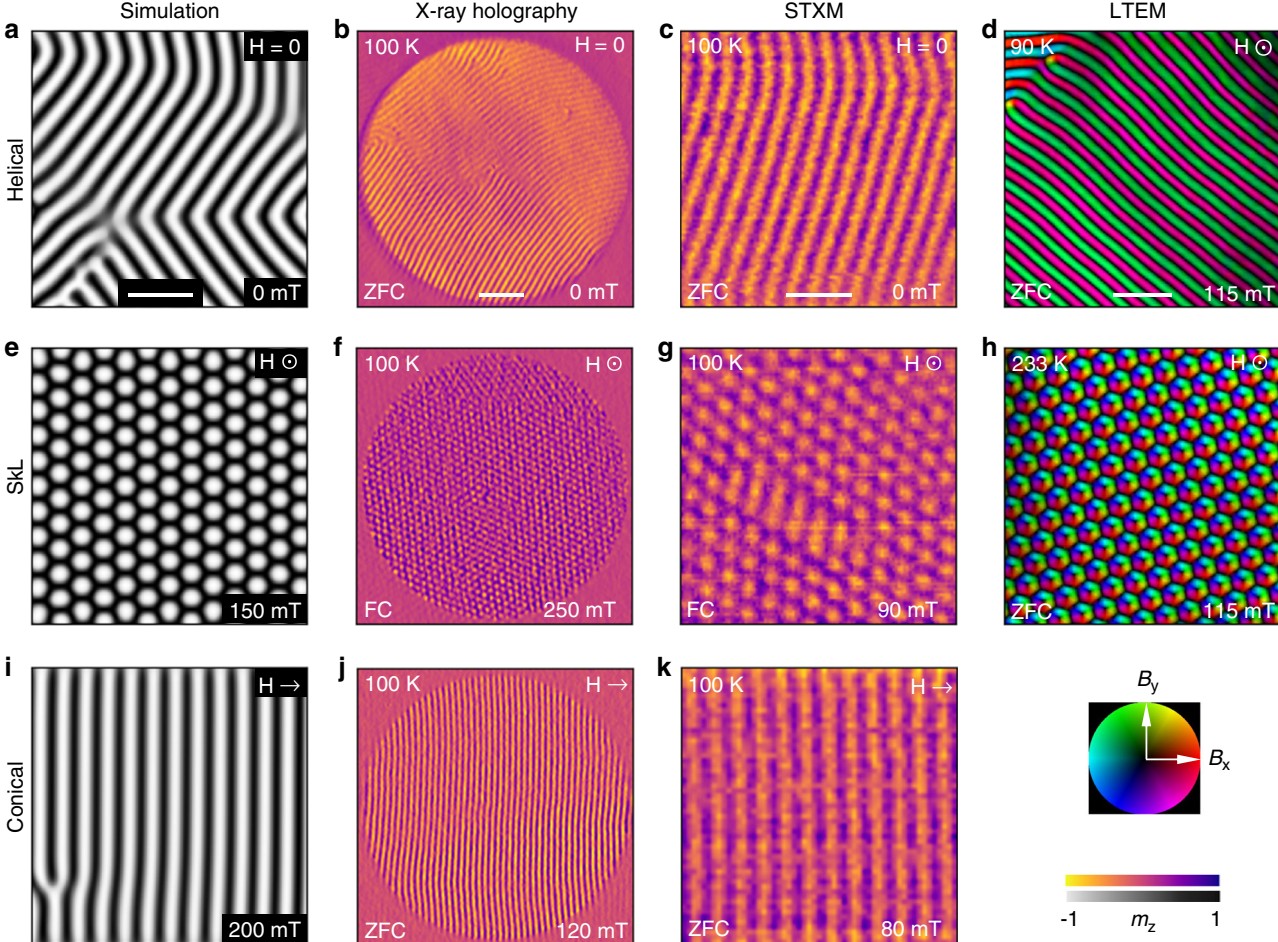

**Fig. 3 Real-space imaging of chiral spin textures.** Micromagnetic simulations, experimental X-ray holography images, STXM images and LTEM images of the **a–d** helical, **e–h** SkL and **i–k** conical magnetic spin textures. The simulation, X-ray holography and STXM images plot the normalised out-of-plane magnetisation, $m_z$, as the colourmap, while the LTEM images plot the in-plane magnetic flux density as the colourmap, with the direction indicated by the colour wheel at the bottom right. Scale bar in **b**, 500 nm. All other scale bars, 200 nm.

To validate the identification of these structures as the SkT state, we performed supporting micromagnetic simulations. These simulations were based on an idealised model, which does not consider surface roughness, sample defects or temperature, but nevertheless provide an invaluable comparison to the experimental data. The simulation was initialised by relaxing a state consisting of three paraboloid-shaped skyrmion tube precursors at a range of in-plane magnetic fields, with the state at 150 mT showing the closest agreement to experiment (see Supplementary Fig. 9 and Supplementary Note 7). The magnetic field was then varied to explore the field-dependent behaviour of the simulated SkT state. The average $m_z$ through the thickness of the simulated sample was calculated to produce a simulated X-ray image of the magnetic state (see Supplementary Figs. 5 and 8, and Supplementary Note 6). Selected simulated images are displayed in Fig. 4e–h, showing remarkable agreement to the corresponding experimental micrographs. In Fig. 4e, at 90 mT, the end of the uppermost skyrmion tube curves into the conical state, while the two lower tubes appear to bulge at the end, replicating the behaviour observed in the experimental image Fig. 4a.

A cross section through the end of one of these simulated tubes is shown in Fig. 4i, highlighting the presence of a magnetic Bloch point. The discretisation of the magnetic spin texture in the simulations means that the estimated energy of large scale objects, such as the skyrmion tube itself, is robust, but small scale objects where the magnetisation rapidly changes, such as the Bloch

points, may be inaccurate. Additionally, the 4 nm cell size in the simulation limited the size of the Bloch points to a nanometer scale, while in reality a Bloch point can be expected to exist on the scale of individual spins—beyond the limits of our current imaging resolution. Nevertheless, the image in Fig. 4a may represent the experimental observation of the magnetic configuration around the Bloch point at the end of a skyrmion tube, and is a crucial first step towards direct experimental comparison to theoretical work on Bloch points[18,35].

Selected three dimensional visualisations of the simulations are displayed in Fig. 4j–m. The additional surface structures, which disappear with increasing field in Fig. 4j–l, are chiral edge twists in the conical state at the sample boundary[36]. At decreasing magnetic fields, the skyrmion tubes branch into the helical state and expand to touch the surfaces of the simulated sample, establishing partial skyrmion tube edge states shown in Fig. 4j. Upon increasing the magnetic field, the skyrmion tubes decrease in length, as seen in the experimental images. Despite the qualitative agreement of the experimental and simulated images exhibited in Fig. 4, we note that in the simulation the SkT state exists over a higher magnetic field range in comparison to the experimental observations. This has the secondary effect of altering the relative magnetic contrast of the SkT and conical structures, due to the reduction of the spin canting angle in the cone state with changing applied field. However, this can be attributed to two factors. Firstly, micromagnetic simulations are

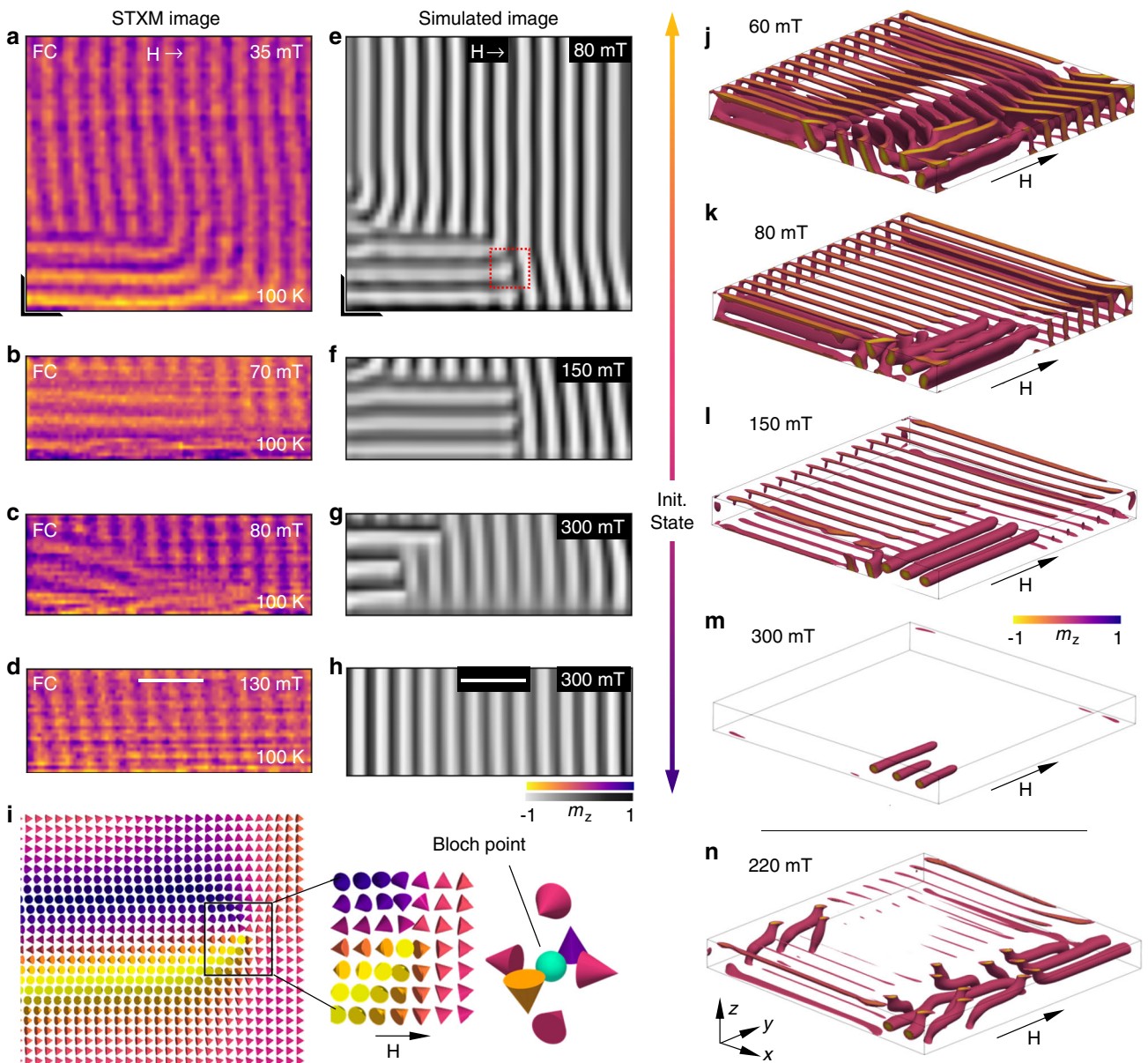

**Fig. 4 Experimental observation and micromagnetic simulations of skyrmion tubes. a–d** Scanning transmission x-ray micrographs of the skyrmion tube spin texture embedded in the conical state observed as a function of applied in-plane magnetic field. The colourmap plots the normalised out-of-plane magnetisation averaged through the thickness of the sample, $m_z$. The black L-shape in **a** indicates the location of the corner of the sample. **e–h** Simulated images of the skyrmion tube spin texture embedded in the conical state, as determined from micromagnetic simulations calculated as a function of applied in-plane magnetic field, where the colourmap plots the normalised out-of-plane magnetisation averaged through the thickness. **i** A cross section of spins from the simulation in **e**, location shown by the red box, highlighting the presence of a magnetic Bloch point at the end of each skyrmion tube. **j–m** Three dimensional visualisations of the micromagnetic simulations for selected magnetic fields, obtained by plotting cells with normalised $m_y$ between -1 and 0. **n** Three dimensional visualisation of the skyrmion tube state achieved after a field sweep from an initially randomised state. Scale bars, 200 nm.

not able to incorporate thermal effects, and are effectively performed at 0 K, whereas the experimental images were acquired at 100 K. Secondly, the lateral extent of the simulated sample is smaller than the measured lamella, modifying the effects of the demagnetising field.

We performed additional simulations commencing from a randomly initialised helical state at 0 mT, and found that the SkT state was also stabilised during an in-plane magnetic field sweep, as depicted by the visualisation in Fig. 4n (see Supplementary Figs. 10 and 11, and Supplementary Notes 7 and 8). In contrast to the previous simulations, the ends of the tubes curve to touch the upper and lower faces of the sample. Such edge states may be energetically favourable in comparison to the formation of a

magnetic Bloch point. Previous studies have demonstrated that the SkL state has improved stability at the sample boundaries for out-of-plane magnetic fields[37]. Our results suggest that the stability of the SkT state is similarly enhanced at the sample edge for in-plane magnetic fields. This may also explain why no SkT state was observed in the in-plane magnetic phase diagram in Fig. 2b, where the field of view was restricted to the centre of the FeGe lamella.

## Discussion

The helical/conical period, $d_{h,c}$, and the distance between each skyrmion tube, $d_{ss}$, were extracted from the experimental and

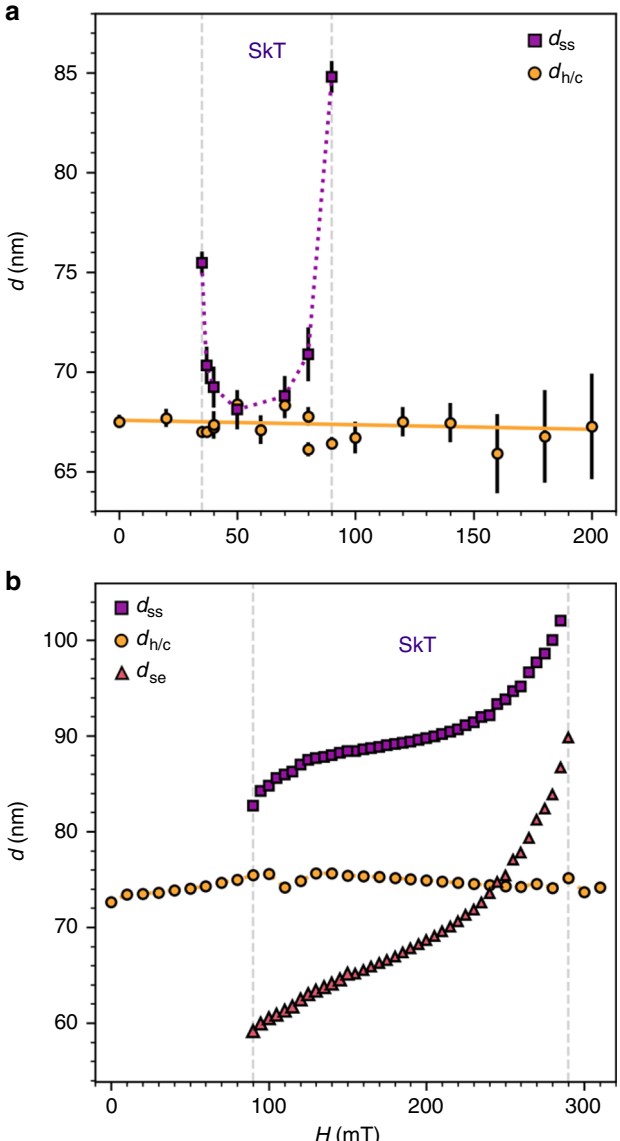

**Fig. 5 Spacing of the skyrmion tube state. a, b** The skyrmion-skyrmion tube spacing, $d_{ss}$, the skyrmion-edge distance $d_{se}$, and helical/conical state period, $d_{h/c}$, plotted as a function of applied magnetic field for the experimental and simulated images respectively. Error bars on the experimental data indicate the standard deviation obtained when fitting line profiles of the magnetic contrast from the STXM images with a sine function.

simulated data, and are plotted as a function of applied magnetic field in Fig. 5a, b, respectively. The skyrmion-edge distance $d_{se}$ for the simulated data is also plotted. In both the simulation and experiment, $d_{h,c}$ was found to remain approximately constant across the full range of applied magnetic field. In contrast, the simulated $d_{ss}$ and $d_{se}$ increase with applied magnetic field, exhibiting skyrmion-skyrmion and skyrmion-edge interactions, which are attractive at low fields, and repulsive at higher fields (see Supplementary Fig. 12 and Supplementary Note 9 for details), in agreement with studies of such interactions for the out-of-plane SkL state[38,39]. At low and high fields, $d_{ss}$ becomes, respectively, smaller and larger than the skyrmion radius. In both instances, the magnetisation of the SkT spin texture necessarily distorts, in a manner functionally similar to the distortion of the helical state into a chiral soliton lattice[40] (see Supplementary Fig. 13 and Supplementary Note 10).

While the experimental behaviour of $d_{ss}$ is not exactly replicated by the simulations, its value is nevertheless highly variable, ranging from 67 to 86 nm, in contrast to the constant value of $d_{h,c}$. This provides strong evidence that the observed SkT spin texture is distinct from these topologically trivial magnetic states. We suggest that the discrepancy may be due to the formation of the SkT structure by field cooling, producing a pinned metastable state which relaxes over the first few field increments, before displaying the expected increase in $d_{ss}$ at higher applied fields. Such pinning effects are not considered in the simulation model. In the future, achieving a lower temperature in the STXM instrument would allow the metastable skyrmion tubes to exist over a larger range of applied field, allowing this behaviour to be examined more thoroughly.

In conclusion, we have demonstrated direct imaging of magnetic skyrmion tubes utilising X-ray microscopy and comparative micromagnetic simulations. This observation confirms, in real-space, the extended nature of the magnetic skyrmion along the applied magnetic field direction. The field-evolution of the SkT state, and its location at the sample boundary in both the experiment and simulation, highlights the importance of confinement and boundary effects in the formation of this magnetic state and the emergent skyrmion-skyrmion and skyrmion-edge interactions. Experimental realisation of in-plane skyrmion tube structures opens the door to further studies of this unexplored dimension of the skyrmion spin texture and its associated dynamic phenomena.

## Methods

**Sample preparation.** Single crystals of FeGe were grown by the chemical vapour transport technique. Two grams of prepared FeGe powder and 2 mg/cc of the iodine transporting agent were used for the growth, with the source maintained at 450 °C and a temperature gradient of 50 °C across the length of the tube, over a period of 1–2 weeks. Several single crystals with dimensions of $1.5 \times 1.5 \times 1.5$ mm³ were obtained at the colder end. From one single crystal crystal, two lamellae, of thickness ~120 nm, with the [112] crystal direction as the plane-normal direction, were prepared via an in-situ lift-out method using a focused Ga ion beam system (FEI Helios Nanolab). Using a in-situ micromanipulator, one lamella was fixed by Pt deposition over a 3 μm aperture in a $Si_3N_4$ membrane coated with 600 nm of Au for the magnetic diffraction measurements. A reference slit of width ~20 nm and length 6 μm was cut 3.5 μm from the centre of the sample aperture, again using the focused ion beam, for X-ray holography measurements. The second lamella, also ~120 nm thick, was attached to a standard Cu TEM grid with Pt deposition, and ion-milled into an L-shape for STXM measurements (see Supplementary Fig. 1 and Supplementary Note 1). For the final ion milling pass, we utilised a low Ga acceleration voltage of 5 keV and a current of 21 pA. Utilising the widely available TRIM software, we estimated that the Ga implantation in the resulting samples was less than ~10 nm, leaving the majority of the lamella unaltered from the bulk crystal.

**Magnetic X-ray diffraction.** Resonant magnetic X-ray diffraction measurements were performed with the RASOR diffractometer at Diamond Light Source, and the COMET instrument at Synchrotron SOLEIL. Sample cooling was achieved by a He cryostat and the applied magnetic field was controlled by varying the arrangement of four permanent magnets. With the instrument setup for transmission experiments, the coherent X-ray beam was directed through the sample aperture and reference slit, and the resultant diffraction pattern captured by a CCD placed downstream of the sample. The magnetic signal was maximised by tuning the X-ray energy to the $L_3$ Fe absorption edge, at ~708 eV after measuring an X-ray magnetic circular dichroism (XMCD) spectrum (see Supplementary Fig. 2 for a schematic and details).

**X-ray holography.** Using the same experimental setup as in the magnetic X-ray diffraction measurements, magnetic X-ray holography was performed at Synchrotron SOLEIL[41]. The transverse coherence length of the SEXTANTS beamline was measured to be 25 μm in both vertical and horizontal directions. We estimated that the flux incident on the sample aperture was $10^9$ photons per second. While standard holography techniques utilise a circular aperture for both the sample and reference apertures, we utilised an extended reference slit, which offers both higher transmitted flux and improved spatial resolution. For each holographic image, two diffraction patterns were recorded with opposite circular polarisation of the incident X-ray beam. For a high quality image, we set the detector distance to 24 cm, and recorded over 100 exposures in each polarisation, for a total measurement time of 3 h. This was primarily limited by the long CCD readout time of a few seconds in

comparison to each 200 ms exposure time. Holographic reconstruction of the magnetic sample image was then performed using the HERALDO technique[42]. We subtracted the two exposures to eliminate structural information and applied a linear differential filter to account for the additional convolution signature from the extended reference slit. Finally, the result was Fourier transformed to acquire the reconstructed image (see Supplementary Fig. 2 and Supplementary Note 2). We estimated the spatial resolution of the presented X-ray holography images to be 25 nm, which was primarily limited by the pixel size on the CCD detector.

**Scanning transmission X-ray microscopy.** Scanning transmission microscopy measurements were performed at the MAXYMUS instrument at BESSY II. With the sample mounted inside the microscope, cooling was achieved by a He cryostat and the applied magnetic field was controlled by varying the arrangement of four permanent magnets. The vibrations from the cryostat were reduced by setting the gas flow to the lowest level feasible to ensure successful low temperature imaging of the sample. The X-ray beam was focused to a 22 nm spot size using a Fresnel zone plate and order separation aperture. This focused beam, once again with an X-ray energy of 708 eV, was then rastered across the sample pixel by pixel using piezoelectric motors. By exploiting the effects of XMCD at the resonant X-ray energy, the transmission of the sample at each point was measured to form an image of the magnetic contrast (see Supplementary Fig. 2 and Supplementary Note 2). The presented images were recorded using a single X-ray polarisation. Background contrast caused by a slight thickness gradient over each image was subtracted (see Supplementary Fig. 6). Transmission of the beam through the sample was measured by an avalanche photo diode with a 2 GHz signal bandwidth. A gated detection with a 20 ps long measurement window was utilised such that the measurement was only active at the expected arrival time of the photons, effectively realising a 500 MHz lock-in on the synchrotron beam pulses. This signal was then compared to a reference voltage to achieve fast single photon counting. A typical image, like those shown in Fig. 3, required an acquisition time of 15 min. We estimated that the spatial resolution of the presented STXM images to be 18 nm, which was primarily limited by the width of the focused X-ray beam[43,44].

**Lorentz electron transmission microscopy.** LTEM measurements were performed on a comparable FeGe lamella using an FEI Tecnai F20 transmission electron microscope operated at an acceleration voltage of 200 kV and equipped with a field-emission electron gun. Images were acquired using a Gatan imaging filter and recorded on a 1024 × 1024 pixel CCD. Pairs of images with equal and opposite defoci were acquired at each temperature and the projected magnetic flux density was calculated from these using the transport of intensity equations.

**Micromagnetic simulations.** Simulations of the various magnetic configurations observed in the experiments were performed using the micromagnetic code OOMMF[45] and the data were processed using the OOMMFPy library, available online[46]. The simulated system was specified with dimensions 1000 × 1000 × 100 nm, using finite difference cells with a volume of 4 nm³, and magnetic parameters of FeGe. We describe the FeGe system using the energy functional of a chiral magnet with symmetry class $T$, which reads

$$E = \int_V dV \left\{ A \sum_{\alpha=x,y,z} (\nabla m_\alpha)^2 + D\mathbf{m} \cdot (\nabla \times \mathbf{m}) - M_s \mathbf{m} \cdot \mathbf{B}_a - \frac{M_s}{2} \mathbf{m} \cdot \mathbf{B}_d \right\}, \quad (1)$$

where $\mathbf{m}$ is the normalised magnetisation, $A = 8.78$ pJm$^{-1}$ is the exchange constant, $M_s = 384$ kAm$^{-1}$ is the saturation magnetisation, $D = 1.58$ mJm$^{-2}$ is the DMI constant, $\mathbf{B}_a$ is the applied field and $\mathbf{B}_d$ is the demagnetising field. The energy minimisation of a specified initial state was performed using OOMMF's conjugate gradient method. The same minimisation technique was applied to reach equilibrium states at each step of the simulated field sweeps (see Supplementary Note 6).

## Data availability
Experimental and simulation data, and the relevant analysis scripts utilised to produce the presented figures are available from an online repository[47]. Further material is available from the corresponding author upon reasonable request.

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

## Acknowledgements

We give thanks for the assistance of M. Sussmuth at Diamond Light Source. We acknowledge the support of Diamond Light Source for time on beamline I10 under proposal SI20866-2, as well as experiment time at SOLEIL (proposal 20180679) and BESSY II (proposal 181-06589ST). This work was financially supported by two Engineering and Physical Sciences Research Council grants: EP/M028771/1, and the UK Skyrmion Project Grant, EP/N032128/1. M.N.W. acknowledges the support of The Natural Sciences and Engineering Research Council of Canada (NSERC).

## Author contributions

M.T.B., L.A.T., M.N.W. and P.D.H. conceived the project and designed the experiments. J.A.T.V. and G.B. produced the FeGe single crystal. M.T.B., L.A.T., N.B. and A.C.T.-H. manufactured the lamella samples. M.T.B., L.A.T., S.H.M., R.F. and P.S. performed the magnetic diffraction measurements at Diamond Light Source. M.T.B., L.A.T., A.L., H.P. and F.Y.O. performed the magnetic holography experiments at SOLEIL. M.T.B., L.A.T., M.N.W., F.G., N.T., M.W. and J.G. performed the STXM measurements at BESSY II. J.C.L. and A.C.T.-H. acquired the LTEM data. D.C.-O., O.H. and H.F. performed the micro-magnetic simulations. M.T.B. and D.C.-O. wrote the paper. All authors discussed the results and commented on the manuscript.

## Competing interests

The authors declare no competing interests.
