## [Peer Review File · Nature Communications]

Reviewers' comments:

Reviewer #1 (Remarks to the Author):

By using resonant magnetic x-ray imaging and comparative micromagnetic simulations, the authors try to observe skyrmion tubes in a lamella of FeGe. Their experimental techniques are high level and they actually observed interesting features. However, there are several questionable data presentations and their interpretations for the demonstration of the first real-space observation of skyrmion tubes, especially about Fig. 4. Main questionable points are as follows.

1) In sample preparation, the authors used focused ion beam to get the L-shaped sample as shown in Fig. S1(d). Skyrmion tubes were found at the left-bottom corner of the thin sample. Even in the red box region, the sample thickness is thinner in the left region near the edge. Thus, radiation damages due to the ion beam are expected to be highest at the left-lower region.

Did the authors estimate the ion damage layers in both top and bottom surfaces? The structure and property of the region containing these damaged regions seem to be largely different from those of the bulk FeGe.

2) In Fig. 4(a), the corner of the sample is indicated by black L-shape, while in Fig. S4(a), there are some image contrasts outside the sample edge indicated by white dashed lines. What do these contrasts come from?

3) In the simulated image of Fig. 4(f), the red box indicates the presence of magnetic Bloch point. On the other hand, there is no such regions in the experimental images in Fig. 4(b) and other figures. Thus, the existence of the Bloch point is not proved with the present experiment.

4) In the experimental images of Fig. 4(a), skyrmion tube like contrast (3 white horizontal lines) are connected to the white vertical lines (conical state), being sharply different from the simulated images of Fig. 4(e), where 2 white lines in the lower part are disconnected from the white lines (conical state).

5) In left-lower region of Fig. 4(c), the number of curved skyrmion tubes seems to be 4 not 3. The contrast of experimental data is quite different from the simulation.

Although the experiments were performed at high level, the data obtained and their interpretations contain the above questionable points, and thus the present study could not prove the demonstration of the first real-space observation of skyrmion tubes.

Reviewer #2 (Remarks to the Author):

The authors report on a Real-space imaging of confined magnetic skyrmion tubes, studied by means of soft X-ray magnetic imaging.

The main novelty of the manuscript consists in the direct observation, by using X-ray magnetic imaging, of skyrmion tubes in a thin lamella of an FeGe crystal and their annihilation in the conical phase when the external magnetic field is increased to a sufficiently high value.

The experimental data presented is the result of challenging experiments with the use of scanning transmission X-ray microscopy at relatively low temperatures.

The existence of these tube-like magnetic structures was already established in skyrmion hosting materials, however I believe that the possibility to observe them in real space is a significant step forward that will help in improving our understanding of the formation and stability of magnetic topological defects.

While diffraction based techniques offer an average picture of the sample "microstructure", direct space imaging offers the opportunity to observe configurations which deviate from the average-like picture. Therefore, I'm convinced that the manuscript will be relevant to scientists working in the specific field as well as those interested in topological (magnetic) defects.

Few comments on the paper, which is in general well written:

1) What is the spatial resolution of the images shown in Fig.3 ? How much is the resolution affected by the magnetic contrast?

Some of these info should appear in the main text and/or in the SI.

2) I think for the reader is not apparent what is the source of the magnetic contrast (m_z component in the paper reference system) observed in the X-rays experiment, when the external magnetic field (and the skyrmion tubes) is applied along the y-direction.

3) I think the authors should discuss their results in connection with the observation/prediction reported in Science 340, 1076 (2013).

In this paper it was suggested by simulation that the merge of two skyrmion tube would occur through the "zipping" of topological magnetic defect across the sample.

The simulations that support the current manuscript seems to imply a different scenario (at least looking at Fig 4g) and

to be honest if I compare Fig 4c and 4g I'm not very convinced that the simulation are correctly reproducing the experimental data.

4) I would recommend the authors to share on Zenodo, as they do for the simulation, also the original data (when applicable) which is shown in Fig 3, 4 and 5.

I hope that these comments will be taken into account to improve an already very nice manuscript.

Methods:

The authors should include in the Magnetic X-Ray Diffraction and holography description if they used a pinhole (and its size, its distance to the sample) and the sample-detector distance.

Do they have an estimate of the coherent flux of the beamline in the experimental conditions? What is the transverse coherence length?

STXM: which detector was used? Can they be more precise on the way "the vibration of the cryostat where reduced?"

How long is the acquisition time to obtain an image as those shown in Fig. 3 (for each technique)?

We would like to thank both referees for taking the time to read our manuscript and for providing their detailed comments and feedback, which we believe have helped improve the manuscript. We respond to the individual points raised below.

Reviewer#1

1) “In sample preparation, the authors used focused ion beam to get the L-shaped sample as shown in Fig. S1(d). Skyrmion tubes were found at the left-bottom corner of the thin sample. Even in the red box region, the sample thickness is thinner in the left region near the edge. Thus, radiation damages due to the ion beam are expected to be highest at the left-lower region. Did the authors estimate the ion damage layers in both top and bottom surfaces? The structure and property of the region containing these damaged regions seem to be largely different from those of the bulk FeGe.”

Firstly, we admit that we did not fully explain the meaning of the red box on the L-shaped lamella image in Fig. S1(d). We believe the reviewer may have interpreted the red box as the spatial extent of the image displaying skyrmion tubes in Fig. 4(a), leading them to suggest that the thickness is greatly varying across the STXM image. However, in reality the size of the image in Fig. 4(a) is smaller relative to S1(d), such that the thickness of the sample can be considered to be roughly constant across Fig. 4(a-d), with only minor variation. We have updated Fig. S1(d) by replacing the red box with a red indication arrow (see Supplementary Information page 3).

We have performed a Ga ion implantation simulation using the widely-available and standard TRIM software. Utilising a highly pessimistic normal incidence angle, we calculated that the Ga ions could have implanted no deeper than 10 nm into the top and bottom surfaces of the sample during fabrication. This is primarily due to the low ion acceleration voltage of 5 kV we utilised when performing the ion milling. Since the sample is 120 nm thick at the thinner edge of the sample, there remains a significant portion of the lamella which is unaltered from the bulk crystalline state. Therefore, any major differences in the magnetic behaviour of the sample compared to bulk FeGe are due to confinement effects such as shape anisotropy, which are widely known, and we do not expect any significant effects from Ga implantation. Details of the milling parameters used in the FIB fabrication and a discussion of the Ga implantation calculation have been included in the methods section (see Main Text line 200).

2) “In Fig. 4(a), the corner of the sample is indicated by black L-shape, while in Fig. S4(a), there are some image contrasts outside the sample edge indicated by white dashed lines. What do these contrasts come from?”

We agree that our explanation of this matter could be improved. In our original Fig. S4(a), we labelled the region where magnetic ordering is no longer observed as the sample edge. This was described in the supplementary information text. However, in the new version of Fig. S4 we have clarified this by including a wider view of the corner of the sample (see Supplementary Information page 11). This illustrates the distinction between the structural edge of the sample and the limit of the observed ordered magnetic contrast.

Such a region is sometimes seen in LTEM images of FeGe lamellae: the magnetic contrast does not extend to the very edge of the sample. This edge region can be composed of redeposited or damaged FeGe which builds up at the edge of the sample during ion milling – a phenomenon which commonly occurs during the FIB fabrication process.

Therefore, we believe the contrast seen in this edge region is structural contrast. The contrast in this region is also somewhat affected by some minor image processing which we performed to

most clearly display the magnetic contrast in each image. The purpose of this processing is to remove the background caused by the slightly varying thickness, and therefore x-ray transmission, across the sample. This contribution appears significant due to relatively small magnetic contrast signal achieved in the STXM images. We have now updated the supplementary information to include a new section with a comprehensive description of this image processing, as required by Nature's editorial policy on microscopy images. (see Supplementary information line 55).

Comments 3), 4) and 5).

Before responding to each of these comments separately, we wish to stress that while the micromagnetic simulations are a valuable qualitative comparison to the experimental data, they cannot seek to fully reproduce the experimental data. The simulations are based on an idealised model which does not consider surface roughness, sample defects or temperature. While the determination of large scale magnetic structures, such as the skyrmion tube, is robust, the discretisation of the magnetic texture in the simulation means that energy estimation of small regions of rapidly varying magnetisation, such as Bloch points, is challenging.

One further issue to consider when comparing the experimental and simulated images is that the spatial resolution of the originally presented simulated images was far greater (cell size of 4 nm) than in the experimental images (~20 nm). Therefore, expecting to identify the same level of detail in the experimental images is unrealistic. We have now altered the resolution of the simulated image panels in Fig. 4 to match the experimental resolution. We have included a representative figure below which depicts the experimental and the simulated tube states plotted at the same spatial resolution. We have also plotted each image in the reverse colour map for comparison. We achieved this by averaging each 5 x 5 array of the 4 nm pixels in the simulated image into a 20 nm pixel. We believe this is a more realistic way to visually compare the images. We have included a new Supplementary Information section discussing these matters and presenting the simulated images at their original resolution.

3) “In the simulated image of Fig. 4(f), the red box indicates the presence of magnetic Bloch point. On the other hand, there is no such regions in the experimental images in Fig. 4(b) and other figures. Thus, the existence of the Bloch point is not proved with the present experiment.”

As stated above, the discretisation of the spin texture in the micromagnetic simulations means that the estimated energy of small scale structures such as the Bloch points is likely inaccurate. As a further consequence of this, the spatial size of the Bloch point is greatly exaggerated in our simulations, where we utilised a 4 nm cell size. In reality, the Bloch points can be expected to exist on the scale of individual spins, which is far beyond the spatial resolution limit of our STXM imaging method. It is therefore expected that we are unable to resolve the Bloch point itself. We have included additional detail in the main text to discuss these issues (see Main Text lines 110, 123).

Accurately modelling such a Bloch point structure is a wider problem in the field beyond our own work. Indeed, we have recently seen a surge of papers, largely presenting simulation work, discussing the three dimensional nature of skyrmion tubes and their associated Bloch points. We believe that our results, and further experimental imaging studies of skyrmion tube states which will follow, will be vital for comparison to such theoretical studies.

In the manuscript, we were seeking to argue that we observe the magnetic configuration of the skyrmion tube surrounding the Bloch point. We suggest that spin-polarised scanning transmission microscopy, which is capable of true atomic resolution, may be capable of resolving a Bloch point. We have reworded this sentence to clarify this in the manuscript, and have now included a brief discussion of the limitations which must be overcome to resolve and simulate the Bloch point structure itself (see Main Text line 126).

4) “In the experimental images of Fig. 4(a), skyrmion tube like contrast (3 white horizontal lines) are connected to the white vertical lines (conical state), being sharply different from the simulated images of Fig. 4(e), where 2 white lines in the lower part are disconnected from the white lines (conical state).”

Firstly, we believe that we may not have fully explained the signature of a skyrmion tube in our experimental data. The reviewer suggested that the skyrmion tubes are the horizontal lines of light contrast, whereas in reality they are both the light and dark lines of contrast. On one side of the skyrmion tube the spins point towards the observer, while on other side they point away from the observer, such that a single skyrmion tube exhibits both light and dark contrast. The second reviewer also identified this as a weak point in our explanation, and we have endeavoured to add additional detail to the manuscript to rectify this (see Main Text line 98).

We now turn to the reviewers comment about the connection of the white horizontal lines to the white lines of the conical state. We would like to highlight simulated images of skyrmion tubes displayed in supplementary Fig. S7, which we have reproduced below for convenience. These simulations display skyrmion tube structures where both the black and white contrast connect to the black and white contrast of the surrounding conical state. Therefore, we believe that such features in our experimental images in Fig. 4 are consistent with the signature expected from a skyrmion tube state.

As a further complication, one subtlety that we previously did not mention is that in the simulations, there can be differences in the local magnetic structure around where the Bloch point meets the conical state. For example, in the first two simulations below, the Bloch point of the skyrmion tube sits at the boundary of the white and black contrast of the conical state. However, in the third simulation, we can see that the presence of the Bloch point has significantly distorted the structure

of the surrounding conical state, such that it no longer follows a straight vertical line. This leads to more obvious connection of both the black and white contrast of the skyrmion tube to the conical contrast. Due to the previously mentioned limitations of current micromagnetic methods, it is difficult to accurately simulate the energy of the Bloch point, and thus the surrounding magnetic structure, and the observed contrast, might be expected to be somewhat different in the experimental data.

5) “In left-lower region of Fig. 4(c), the number of curved skyrmion tubes seems to be 4 not 3. The contrast of experimental data is quite different from the simulation.”

We agree that there appear to be 4 lines of light horizontal contrast in the original version of Fig. 4(c). However, as stated above, the signature for a skyrmion tube is in fact a line of both light and dark contrast. We believe that this additional small area of white contrast is structural contrast from the edge of the sample. We have reanalysed the data in Fig. 4 and refined the processes we performed on each image. These processes are now fully described and demonstrated in the supplementary information (see Supplementary Information line 55 onwards). In combination with presenting the simulated images at the same resolution as the experimental images, we believe that Fig. 4 is significantly improved as a result of these changes (see Main Text page 10).

Reviewer #2

1) “What is the spatial resolution of the images shown in Fig.3 ? How much is the resolution affected by the magnetic contrast? Some of these info should appear in the main text and/or in the SI.”

In Fig. 3, the spatial extent represented by each pixel for the holographic and STXM images was 25 nm and 15 nm respectively, while the spatial resolution of these images was 25 nm and 18 nm. In the x-ray holography measurements, we were limited by the resolution of the CCD detector itself, while in the STXM measurements, the limiting factor was the width of the focused x-ray beam.

The magnetic contrast itself does not strongly affect the resolution of the images. The magnetic structures of interest all exhibit a roughly sinusoidal profile. We can define a contrast to noise ratio as the maximum magnetic contrast divided by the standard deviation of the experimental noise. This noise value was calculated by taking a line scan over a region of a representative image where there is no expected change in magnetic contrast – such as along a white or black stripe in the helical state. We calculated signal/noise ratios of 9.70 and 8.71 for the holography and STXM experiments respectively.

If we assume that features with contrast less than the image noise cannot be resolved, an upper bound contrast-limited spatial resolution can be estimated. For sinusoidal features with a wavelength of 75 nm the contrast limit was 11 nm for the holographic images and 12 nm for the STXM images. This shows that the limiting factor for resolution is not the magnetic contrast, but rather limitations of the instruments. However, as the magnetic contrast in the image is reduced, for example in the conical state as the applied in-plane magnetic field is increased and the canting angle is reduced, this may become a factor in limiting the image resolution. We have included the resolution limits in the methods section (see Main Text lines 225, 245).

2) “I think for the reader is not apparent what is the source of the magnetic contrast (m_z component in the paper reference system) observed in the X-rays experiment, when the external magnetic field (and the skyrmion tubes) is applied along the y-direction.”

We agree that we could be clearer on this point, specifically to avoid confusion over the contrast exhibited by a skyrmion tube. We have clarified the source of the magnetic contrast in the experimental images, and specifically explained the expected contrast signature of the SkT state (see Main Text line 98).

3) “I think the authors should discuss their results in connection with the observation/prediction reported in Science 340, 1076 (2013). In this paper it was suggested by simulation that the merge of two skyrmion tube would occur through the “zipping” of topological magnetic defect across the sample. The simulations that support the current manuscript seems to imply a different scenario (at least looking at Fig 4g) and to be honest if I compare Fig. 4c and 4g I'm not very convinced that the simulation are correctly reproducing the experimental data.”

We agree that discussion of these skyrmion annihilation dynamics could prove illuminating for the reader. The literature points towards two annihilation mechanisms for the skyrmion state. In the reference suggested by the reviewer, the authors present how the skyrmion state decays into the helical state by forming Bloch points which zip neighbouring skyrmion tubes together.

However, in other works, such as those investigating chiral bobbars, skyrmion tubes are suggested to decay into the conical state by the motion of the Bloch points at their ends. This is the mechanism

suggested by the data and simulations in our current paper. It is likely that the exact microscopic mechanism depends on the specific orientation of the magnetic state to which the skyrmions are transitioning. Data from future imaging experiments and those presented in this paper will be a vital comparison to such theoretical models. We have included some discussion of different annihilation mechanisms in the introduction of our paper to discuss this point (see Main Text line 30).

In reference to the agreement of the experimental data with the simulations, we wish to stress as we did in response to Reviewer #1 that while the micromagnetic simulations are a valuable qualitative comparison to the experimental data, they cannot seek to fully reproduce the experimental data. The simulations are based on an idealised model which does not consider surface roughness, sample defects or temperature. While the determination of large scale magnetic structures, such as the skyrmion tube, is robust, the discretisation of the magnetic texture in the simulation means that energy estimation of small regions of rapidly varying magnetisation, such as Bloch points, is challenging. Therefore, exact reproduction of the experimental data is unrealistic. We have included some of these arguments in the manuscript (see Main Text lines 110, 123).

4) “I would recommend the authors to share on Zenodo, as they do for the simulation, also the original data (when applicable) which is shown in Fig 3, 4 and 5.”

We recognise that the sharing of our experimental data is crucial for full transparency of our results. We have uploaded python analysis scripts used to create figures 3, 4 and 5, and the associated raw data into both an Zenodo and GitHub repository: <https://github.com/davidcortesortuno/paper-2020-real-space-imaging-of-confined-magnetic-skyrmion-tubes>. By clicking the “launch binder” badge, a copy of the data repository is created, and the associated scripts can then be run and edited directly in the cloud via web browser.

5) Methods Section Comments:

“The authors should include in the Magnetic X-Ray Diffraction and holography description if they used a pinhole (and its size, its distance to the sample) and the sample-detector distance. Do they have an estimate of the coherent flux of the beamline in the experimental conditions? What is the transverse coherence length? STXM: which detector was used? Can they be more precise on the way “the vibration of the cryostat where reduced?” How long it the acquisition time to obtain an image as those shown in Fig. 3 (for each technique)?”

The holographic imaging was performed with an extended reference slit, rather than a pinhole aperture, known as the HERALDO method. We have found through past work that this provides a higher resolution than using a pinhole aperture. However, it does necessitate the use of a differential filter in order to properly reconstruct the image in the final Fourier transform. We have updated the methods section to provide additional detail on the diffraction and holography measurements and subsequent analysis (main text line 213). The geometry of the reference and sample apertures of the holography sample are described in the sample preparation section.

If they are interested in the finer details of the analysis, we would welcome and encourage the reviewer to follow the holographic reconstructions via the python notebooks available in the online data depository.

REVIEWERS' COMMENTS:

Reviewer #1 (Remarks to the Author):

Taking into account the referees' comments, the authors included additional information and revised the manuscript. Just one thing, I noticed that at the specimen edge region the authors tend to attribute some contrast to structural one not magnetic one. In this experiment, is there any way to separate the magnetic information from structural one? This point is important, since there exists more or less mixed state consisting of partly magnetic structure. Thus, I hope that the authors should add some explanations for this point.

Reviewer #2 (Remarks to the Author):

The authors answered satisfactorily to my comments.

We once again thank the reviewers for their detailed feedback and comments from the manuscript. We respond to the final individual points below.

Reviewer#1

1) "I noticed that at the specimen edge region the authors tend to attribute some contrast to structural one not magnetic one. In this experiment, is there any way to separate the magnetic information from structural one? This point is important, since there exists more or less mixed state consisting of partly magnetic structure."

Using STXM, there are a number of ways to attempt separation of magnetic contrast and structural contrast. One could image the sample above and below T_c , thus acquiring the structural information. Another way to acquire this background would be to image the sample at high applied magnetic field, thus imaging only a field-polarised state. A final alternative is to acquire the same image for both left and right x-ray circular polarisation – because only the magnetic contrast reverses, the subtraction of these two images will show solely magnetic information. We attempted to perform the two polarisation subtraction, however, due to sample drift, and the relatively large size of the image pixels, this proved unsuccessful with the data we have available. We have included this discussion in Supplementary Note 4.